# Epidemiology of Carbapenem-Resistant Klebsiella Pneumoniae Co-Producing MBL and OXA-48-Like in a Romanian Tertiary Hospital: A Call to Action

**DOI:** 10.3390/antibiotics14080783

**Published:** 2025-08-01

**Authors:** Violeta Melinte, Maria Adelina Radu, Maria Cristina Văcăroiu, Luminița Mîrzan, Tiberiu Sebastian Holban, Bogdan Vasile Ileanu, Ioana Miriana Cismaru, Valeriu Gheorghiță

**Affiliations:** 1Department of Infectious Diseases, “Carol Davila” University of Medicine and Pharmacy, 050474 Bucharest, Romania; maria-adelina.cosma@drd.umfcd.ro (M.A.R.); valeriu.gheorghita@umfcd.ro (V.G.); 2“Agrippa Ionescu” Clinical Emergency Hospital, 050474 Bucharest, Romania; vacaroiu.cristina@dcti.ro (M.C.V.); holban.tiberiu@dcti.ro (T.S.H.); miriana.ioana@yahoo.com (I.M.C.); 3Center for Health Outcomes and Evaluation, 050474 Bucharest, Romania; bogdan.ileanu@csie.ase.ro; 4Faculty of Cybernetics, Statistics and Informatics, University of Economic Studies, 050474 Bucharest, Romania

**Keywords:** Carbapenem-resistant, *Klebsiella pneumoniae*, metallo-β-lactamases, NDM/OXA-48-like carbapenemases, cefiderocol susceptibility, antimicrobial stewardship

## Abstract

**Introduction**: Carbapenem-resistant *Klebsiella pneumoniae* (CRKP) represents a critical public health threat due to its rapid nosocomial dissemination, limited therapeutic options, and elevated mortality rates. This study aimed to characterize the epidemiology, carbapenemase profiles, and antimicrobial susceptibility patterns of CRKP isolates, as well as the clinical features and outcomes observed in infected or colonized patients. **Materials and Methods**: We conducted a retrospective analysis of clinical and microbiological data from patients with CRKP infections or colonization admitted between January 2023 and January 2024. Descriptive statistics were used to assess prevalence, resistance patterns, and patient outcomes. Two binary logistic regression models were applied to identify independent predictors of sepsis and in-hospital mortality. **Results**: Among 89 CRKP isolates, 45 underwent carbapenemase typing. More than half were metallo-β-lactamase (MBL) producers, with 44.4% co-harbouring NDM and OXA-48-like enzymes. Surgical intervention was associated with a significantly lower risk of sepsis (*p* < 0.01) and in-hospital mortality (*p* = 0.045), whereas intensive care unit (ICU) stay was a strong predictor of both outcomes. ICU admission conferred a 10-fold higher risk of sepsis (95%Cl 2.4–41.0) and a 40.8-fold higher risk of in-hospital death (95% Cl 3.5–473.3). **Limitations**: This single-center retrospective study included a limited number of isolates in certain groups. Additionally, cefiderocol (FDC) susceptibility was assessed by disk diffusion rather than by the broth microdilution method. **Conclusions**: Our study underscores the increasing prevalence of metallo-beta-lactamase-producing CRKP, particularly strains harbouring dual carbapenemases. Timely recognition of high-risk patients, combined with the implementation of targeted infection control measures and the integration of novel therapeutic options, is crucial to optimize clinical management and reduce mortality associated with CRKP.

## 1. Introduction

*Klebsiella pneumoniae* (KP) is an opportunistic Gram-negative pathogen that is part of the *Entrobacteriaceae* family. These bacteria are present in a variety of environmental sources, such as domestic and wild animals [1], and antibiotic-resistant strains can easily contaminate fresh fruits, vegetables, meat, and dairy products [2]. KP is particularly responsible for healthcare-associated infections in hospitalised or immunodeficient patients [3]. In addition to causing sepsis, pneumonia, liver infections, urinary tract infections, and wound infections [4], KP can colonise different anatomical sites, increase the risk of sepsis, and considerably raise the probability of KP spreading to contacts.

With a prevalence of 11.7% in acute-care hospitals, KP placed second among isolated bacteria in a recent assessment of healthcare-associated infections (HAIs) in Europe [5]. KP is clinically relevant due to its noteworthy tendency to develop resistance to all kinds of potentially active antibiotics, which have been the mainstay for treating serious infections caused by *Enterobacterales*. The primary cause of resistance in KP is the acquisition of several genes encoding β-lactamases that are able to break down carbapenems, also known as carbapenemases, such as KPC-type (molecular class A, active-site serine), OXA-48-type (molecular class D, active-site serine), and New Delhi metallo-beta-lactamases (NDM) [6,7]. Hypervirulent *K. pneumoniae* (HVKP) strains, which originated from “classical” KP, have become increasingly resistant to a broader spectrum of antibiotics during the past few decades via the acquisition of genes that encode siderophores and improve capsule synthesis [4]. These strains cause mostly community-acquired infections, in both immunocompromised and healthy people [8,9].

Given that the prevalence of carbapenem-resistant *Klebsiella pneumoniae* (CRKP) has reached exceptionally high levels in some countries (e.g., 72% in Greece, 47.3% in Bulgaria, 41.87% in Russia, 35.29% in Brazil, and 24.6% in the USA) [10], it has been added to the WHO priority list of key resistant infections for the development of novel antibiotics and is currently considered one of the greatest public health challenges [11]. According to a recent meta-analysis, the incidence of CRKP colonisation ranges from 2% to 73%, with a pooled incidence of 22.3%, whereas the prevalence of CRKP colonisation varies globally from 0.13 to 22%, with a pooled prevalence of 5.4% [12]. The three most common classifications for CRKP isolates are extensively drug-resistant (XDR), multidrug-resistant (MDR), and pandrug-resistant (PDR), which makes treating infections much more challenging.

The significance of the molecular epidemiology of CRKP isolates lies in its ability to identify possible therapeutic approaches. Until recently, colistin (COL) was the only available therapeutic option for carbapenem-resistant *Enterobacterales* [13]. Researchers were recently alerted to the nephrotoxicity, unfavourable pharmacokinetic (PK)/pharmacodynamic (PD) profile, and poor effectiveness of polymyxin antibiotics against bacteria that produce class A, B, and D carbapenemases [14], which led to the necessity for the discovery of novel antibiotics. According to the sensitivity profile and carbapenemase type, the therapeutic options for CRKP are limited to a few novel drugs, including ceftazidime/avibactam (CZA), meropenem/vaborbactam (MEV), imipenem/relebactam (IMR), eravacycline (ERV), tigecycline (TGC), cefiderocol (FDC), aztreonam/avibactam (AZA), and some other synergistic combinations. Some novel beta-lactam/beta-lactamase inhibitor (BL/BLI) combinations are currently undergoing phase 3 clinical trials and could become accessible in the coming years as crucial alternatives for treating infections due to CRKP. Of these, the most threatening are metallo-beta-lactamase (MBL)-producing strains [15].

Romania, next to Poland, Italy, Greece, Portugal, Bulgaria, and Cyprus, is endemic for CRKP strains [16]. The rate of CRKP isolates increased from 0% in 2007 to 47.8% in 2022 [10], most likely due to several factors such as suboptimal infection control policies that allowed for horizontal transmission, as well as the inappropriate use and overuse of antimicrobials. Not all hospitals or long-term care facilities in Romania have policies that particularly address antimicrobial resistance. In Romania, we use “Watch” antibiotics as much as “Access” antibiotics, placing us in the top 3 antibiotic consumers in Europe, along with Greece and Cyprus [17].

This study aimed to describe the epidemiological characteristics of the CRKP strains in a multidisciplinary emergency hospital, especially those producing MBLs with or without other carbapenemases. In addition, we sought to determine the characteristics of patients who may be infected and/or colonised with CRKP, as well as to assess susceptibility to available antibiotics and the clinical outcomes of patients infected with these strains.

We also highlight the differences between the breakpoint and the values obtained for CZA, COL, and FDC, along with the potential correlation between this and the beneficial synergism for FDC.

## 2. Results

### 2.1. Main Sample Description

Our sample comprised 71 patients associated with 77 hospital events and 89 bacterial samples. Relative to the number of patients, the prevalence of CRKP was 19.0%. Most of the patients (76.1%, 54/71) were male and above 64 years of age (Table 1). The patient ages were homogenous around the mean, with a standard deviation of 13.8 and a mean of 63.7 years. Among the discharged patients, 72.7% had been healthcare-associated infections and 62.2% had a Charlson score greater than 5. Among all of the hospitalised patients, 22 (31%) developed sepsis. Around two thirds of the patients were admitted to the ICU during their hospital stay, spending a mean of 12 days (s.d. 11.1) and a median of 10.5 days there. CRKP rectal carriers were identified in nearly 20% of the patients, and surgical interventions were performed in about 55.8%. Sepsis occurred in 30.9% of the patients and in 31.2% of hospitalisation events. In 34 episodes, there was a positive screening for rectal carriage, and in 15 cases, sepsis was diagnosed due to the same strain. The patients received empiric or adapted antimicrobials, using at least active drugs in 59.2% of episodes, for approximately 13.6 days on average (s.d. 13.4). Half of the patients received at most 12.2 days of active treatment (Table 1).

The length of stay during hospitalisation varied between 1 and 82 days, with a mean of 23.2 days (s.d. 19.1). Due to such high variation, we should mention that the median length of stay in hospital was 18.0 days. The collection of the 89 samples occurred a mean of 9.1 days (s.d. 12.4) after the first day of admission, with a range of 0 to 61 days and a median of 4 waiting days. The hospital mortality rate among the patients was 22.5% (16/71).

### 2.2. Carbapenemase Type and Susceptibility Profile

We tested 45 of the 89 samples for the presence of carbapenemase-producing strains. The 45 strains were KPC (n = 5), NDM or NDM plus OXA-48-like (n = 24), and OXA-48-like only (n = 16) carbapenemase producers. Our data showed that most samples (53.3%) were positive for NDM or NDM plus OXA-48-like. Of the 24 strains producing NDM or NDM plus OXA-48-like carbapenemase, 20 (83.3%) were co-producers of NDM and OXA-48-like. Therefore, 44.4% of the tested samples were double carbapenemase producers.

The susceptibility profile for our group of patients revealed high rates of resistance to TGC, IMR, fosfomycin (FOS), and amikacin (AMK), of 92.7%, 91.7%, 80.3%, and 76.4%, respectively, restricting the available treatments. According to the sensitivity profile, the most active antibiotics, based on in vitro data, were FDC, CZA, and COL. The resistance rates to CZA and COL were 45.8% and 56.0%, respectively, out of 83 strains tested for CZA and 84 tested for COL. Among the 46 strains assessed for their response to FDC, 41.3% were resistant. Although the number of isolates tested for ERV was relatively small (n = 16), they showed the highest susceptibility rate (87.5%). Regardless of the type of carbapenemase, TGC had the highest resistance rate (92.7%), in contrast to ERV (Figure 1).

The KPC-producing strains were all resistant to TGC, COL, AMK, and FDC, while maintaining sensitivity to CZA and ERV. Moreover, all strains positive for NDM or NDM plus OXA-48-like were fully resistant to CZA, IMR, AMK, and sulfamethoxazole/trimethoprim (SXT). A larger proportion of CRKP with NDM+OXA-48-like carbapenemases were also resistant to TGC (90.9%), FDC (58.8%), and COL (58.3%). The highest sensitivity in this group was to ERV (97.5%). While the MBL-producing CRKP were CZA-resistant, all OXA-48-like-positive strains remained fully susceptible to CZA (Figure 2 and Figure 3).

All 8 strains resistant to ERV and all 22 strains resistant to CZA were NDM producers or NDM–OXA-48-like co-producers. Except for their responses to ERV and CZA, the tested strains showed different mixed resistance profiles. Specifically, the samples resistant to FDC had 63.2% NDM or NDM–OXA-48-like, 26.3% KPC, and only 10.5% OXA-48-like enzymes, while among the COL-resistant strains, the tests showed 45.2% NDM or NDM plus OXA-48-like, 38.7% OXA-48-like, and 16.1% KPC carbapenemases (Figure 3).

From the analysis of 83 samples for CZA susceptibility, we underline the heterogenous distribution of MIC values in reference to the specific cut-off point of 8 mg/L. On one side of this cut-off, 38 of 83 (45.7%) samples had an MIC of ≥16 mg/L. On the other side, the MIC values for 40 samples (48.2%) clustered around 0.5 and 1. The same large spread characterised the distribution of MIC values for COL and meropenem (MER). In particular, for COL, 41 of 84 cases had an MIC above 16, eight times more than the breakpoint. For MER, our CRKP strains expressed an MIC above 16 in 78 out of 89 samples. Thus, 87.6% of the MER-resistant samples had an MIC eight times higher than the breakpoint. Resistance to FDC, depending on the diameter, revealed a more homogenous distribution. More than two thirds of cases (68.8%) lay within the 18–24 mm interval. The standard deviation was 4 and the mean was 20.1, very close to the cut-off value of 20. Strains having an MIC value close to the breakpoint have a greater risk of therapeutic failure, although in vitro activity might differ from in vivo effects. In this situation, the clinician should thoroughly adapt antibiotic doses, considering both the MIC and specific organ failure in the patient.

Even though the EUCAST breakpoints broadly demonstrated the susceptibility profile, we further analysed the MIC values recorded for these CRKPs. Regarding CZA, the observed MICs were nearly “mirrored” for resistant vs. sensitive strains, with values that were distant from the breakpoint reference (Figure 4). According to the European Committee on Antimicrobial Susceptibility Testing (EUCAST) guidelines for FDC disk diffusion testing, sensitivity and resistance are predictive outside the zone of technical uncertainty (ATU) when the testing is properly performed and calibrated. Given the difficulty of interpreting susceptibility in the area of technical uncertainty (ATU) during routine testing, it is recommended to ignore the ATU and interpret using the zone diameter breakpoints in the breakpoint table [18]. Our data showed that FDC-susceptible strains had a tight range of 23–25 mm, while the resistant profile varied between 13 and 18 mm with no discernible pattern (Figure 4).

### 2.3. Patient Risks and Outcomes

Blood cultures obtained five days after admission were positive twice as often (43.2%) as those obtained before the fourth day after admission (20%). Among all the patients who were discharged with a CRKP infection during the time under analysis, the in-hospital mortality rate was calculated to be 19.5%. Several factors have a significant impact on the accumulated risk of death during a hospital stay. We discovered that 37 cases (48.1%) among the 77 hospital admissions had a positive sample during the first three days of hospitalisation. Among these 37 cases, 16.2% developed sepsis during the period of hospitalisation. From the evaluation of the remaining 40 cases, it emerged that 18 patients (45%) developed sepsis during their hospital stay. According to the Fisher Exact Test results, the difference was statistically significant (*p* < 0.001). Thus, the incidence of sepsis is significantly and positively correlated with healthcare-associated infections (HAIs).

The next two models in our inquiry quantified the impact of the most important available factors on the risks of hospital mortality and sepsis. We are aware that several of the covariates explain each other; Table 2 provides a matrix of associations.

Following the results from Table 2, we identified weak but statistically significant associations of active treatment with surgical intervention (*p* < 0.05) and sepsis (*p* < 0.05). Further, we found a moderate association between ICU stay and the presence of sepsis (*p* < 0.001).

The analysis continued with two logit models that aimed to estimate the cumulative risk of sepsis infection during a hospital stay, along with the risk of mortality. First, we looked at technical indicators. Studying the overall validation measures showed that the HL test pointed to a significant statistical association between the expected and observed probabilities in both risk models (*p* = 0.926 and 0.789). The Nagelkerke R squared values were 0.47 and 0.33, respectively. These values suggest a good capacity for prediction in both risk models. The overall classification proportion suggested that 84.9% of current cases in the sepsis model and 76.7% of cases in the mortality model were correctly classified with these logit models. Through comparison of these values with the recommended threshold of 80%, we underline our models’ good capacity for risk prediction. Although not presented here, one could test the potential inclusion of interaction between sepsis and surgical intervention as an additional covariate in the mortality model. Our results showed that in this scenario, the overall classification accuracy dropped from 84.9% to 79.4%. Also in this context, the Nagelgerke R squared value was diminished to 0.245. In addition, all variables except ICU stay had a non-significant statistical impact on mortality. Therefore, we avoided adding supplementary explanatory variables, and we continued with the models synthesised in Table 3.

Complementarily, the logit model emphasised the adjusted contribution of each factor to the risks of mortality and sepsis. We found that presence in the ICU, regardless of the LoS there, increased the risk of death by 41 times (95% CI 3.5–473.3) in comparison with no ICU admission. Surgical intervention seemed to reduce the probability of a patient’s death during their stay by nearly 95% (95%CI 67–99%) (Table 3).

The cumulative risk of sepsis had three significant determinants. According to the model, surgical intervention reduced the chances of sepsis developing by 74%. This may reflect the importance of source control in managing infections that require removal of the focus of infection. Three crucial elements of source control impact the result: the timeliness of the source control intervention; the approach taken to accomplish source control; and the effectiveness of the source control process in terms of eliminating the infection source and any lingering contamination. The rationale behind source control is that the systemic inflammatory response induced by the infection can be prevented or reduced by directly eliminating the source of infection.

However, hospitalisation increased the chances of sepsis by 3% (95% CI 0–7%) per day, while admission to the ICU caused the chance of sepsis to soar by 10 times (95% CI 2.4–41). Associated comorbidities did not have a significant impact (*p* = 0.617) (Table 3).

## 3. Discussion

Antimicrobial resistance is a major global health concern that decreases the efficacy of available drugs against most common infections. The Organisation for Economic Cooperation and Development (OECD) has projected that by 2035, there will be twice as much resistance to antibiotics used as a last resort as there was in 2005 [19].

### 3.1. CRE (MBL) Epidemiology

The current research indicates that KP isolates are becoming more resistant. Among CRKP clinical isolates, the most prevalent resistance mechanism is the synthesis of carbapenemases [20]. In most cases, carbapenem resistance coexists with aminoglycoside resistance in Gram-negative bacteria. A meta-analysis regarding KP as a nosocomial pathogen in Asia revealed increased prevalence rates of drug resistance against aztreonam, carbapenems, and amikacin, of 73.3%, 65.6%, and 40.8%, respectively [21]. However, there are geographical and temporal variations in the prevalence of carbapenemases. NDM enzymes were predominant among the carbapenemases found in Northern China in the period 2018–2021 [22] and in Egypt in 2018–2019, while in Turkey, OXA-48 and KPC were the most encountered enzymes between 2017 and 2020, accounting for 40.6% each [23]. Guo et al.’s research extended back to 1980 in 105 countries, looking for co-carriers of carbapenemases. The three countries with the highest incidence of double producers of carbapenemases were Thailand (284 strains), the United States (82 strains), and China (78 strains). Thailand and the USA reported NDM/OXA prevalence rates of 100% and 65.9%, respectively, while China reported a prevalence of 50.5% for NDM/KPC-associated carbapenemases [24]. Between 2010 and 2021, the rate of co-carriers increased by 0.4% to 9.67%; these co-carriers threaten even the most recently discovered antimicrobials, like FDC, to which they already show resistance [24,25,26].

A study conducted in Romania between 2015 and 2019, in asymptomatic carriers, showed a rise in the CRKP rate from 1.15% to 5% [27]. Most strains assessed in Romania during 2015–2019 were OXA-48-positive [27,28]. Conversely, our research demonstrated a detrimental shift in the direction of MBL prevailing in 2023. More than half of the tested strains were NDM producers, while 44% of them were double NDM/OXA producers. We can only assume that several factors could have contributed to the increased prevalence of MBL-producing CRKP, such as selection pressure associated with the extensive use of highly active antibiotics on OXA-48-producing bacterial strains, especially CZA; the horizontal transmission of MBL-producing bacterial clones in the context of suboptimal infection control; or plasmid-encoded resistance mechanisms.

Compared to the National Point Prevalence Survey conducted in 2023, in which KP placed second after *Clostridioides difficile* infections, with a prevalence of 10.71% among isolated bacteria involved in HAIs, our study demonstrated a local prevalence of 10.69% CRKP HAIs [29].

### 3.2. Risk Factors

The antimicrobial resistance of KP is associated with several risk factors. As other studies have shown, ICU patients are more susceptible to contracting an XDR-KP infection since they have more serious underlying illnesses and weakened immune systems. Solid tumours and septic shock were found to be independent risk factors for 28-day mortality due to CRKP [30]. Recent use of carbapenem or tigecycline, invasive surgical procedures, and pre-existing digestive system disorders are also risk factors for CRKP according to some studies conducted in China [31,32]. Compared to those for respiratory and intra-abdominal infections, the death rate for CRKP bloodstream infection (BSI) is greater.

In our study, we revealed that surgical intervention decreased the cumulative risk of death during hospitalisation, probably because of the rapid reduction in the bacterial load and better control of the septic process. ICU hospitalisation, regardless of the LoS, was associated with a 40-fold increase in the risk of death compared to patients who did not require ICU admission. Additionally, an ICU stay was associated with a 10-fold increase in sepsis risk during hospital events. Our findings related to ICU factors align with previous results from China [33]. Although referring to various comorbidities as distinct factors, like diabetes mellitus, chronic lung disease, kidney disease, or neoplasm, the authors did not find any association between associated diseases and in-hospital mortality. Our approach quantified associated comorbidities through the Charlson Index. Similarly to other findings we could not reject the null hypothesis of no statistically significant impact (*p* > 0.05) on both the sepsis and mortality risk models [33].

### 3.3. FDC Resistance

The majority of *Enterobacterales* that produce MBL were found to be FDC-resistant, particularly those that were also ceftazidime/avibactam-resistant [34]. While the production of beta-lactamases is the most frequent cause of resistance to broad-spectrum cephalosporins in Enterobacterales, PBP-3 modifications appear to be important nowadays, at least when it comes to novel BLs/BLIs like CZA and AZA, and less so when it comes to IMR [35]. The decrease in outer membrane porin function and SHV (sulfhydryl variable) beta-lactamase activities are also associated with FDC resistance in KP [36]. The combination of FDC and avibactam could be an alternative to boost efficacy against microorganisms that are resistant to FDC.

Treatment of MBL-producing bacteria is difficult because of a potential cross-resistance mechanism between CZA and FDC, as previously reported [37]. Horizontal resistance transmission can be explained by the correlation between FDC resistance and prior hospitalisation.

In our study, 41.3% of the tested strains were non-susceptible to FDC, as determined via the disk diffusion method, while 45.8% and 56% were resistant to CZA and COL, respectively. In contrast to the heterogenous distribution of susceptibility diameters for the strains tested in response to FDC, when COL sensitivity was assessed, 11 strains exhibited CMI values equal to the breakpoint value. Thus, the results should be interpreted with caution; ideally, strains with susceptibility data in the ATU should be retested using the microdilution method.

### 3.4. Carriage/Sepsis

The likelihood of developing a bloodstream infection from carbapenemase-producing Enterobacterales (CPE) carriage outside of high-risk environments is poorly understood. A study including 6828 CPE carriers showed that, compared with KPC, the subhazard of a BSI was lower for NDM and OXA-48-like, but these differences did not reach statistical significance. The subhazard of a BSI was higher among patients with CPE carriage first detected in the ICU or oncology/haematology wards compared with medical wards [38]. In our setting, 15 patients, out of 34 positive screenings for CRKP, developed sepsis with the same strain.

### 3.5. Aztreonam Plus Ceftazidime–Avibactam Synergy

Few options are available when it comes to combating the challenge of continuously increasing antibiotic resistance. COL, TGC, and FOS might be the last resort among the “old” antibiotics [39]. In the past five years, the European Medical Agency (EMA) and the U.S. Food and Drug Administration (FDA) have launched and approved several antimicrobials with varying degrees of effectiveness against MDR Gram-negative bacteria. These are FDC, plazomicin, ERV, and temocillin. Furthermore, IMR, MEV, CZT, and CZA are powerful substitutes for beta-lactam antibiotics [40]. According to our research, the remaining therapeutic alternatives appear to be CZA, COL, and FDC. When confronted with a pandrug-resistant Gram-negative strain, although there are new, more potent antibiotics and combinations of BLs/BLIs, only FDC and the combination of aztreonam (ATM) and avibactam have activity against MBL [41]. CZA and ATM have been successfully used in CRKP infections, in which the synergism between these two was proved [42,43]. AZA is the key combination as a new promising option for MBL-producing CRKP and was approved by EMA on 22 April 2024 [44,45]. Further research is needed to defeat KP’s virulence and evasion mechanisms. Several promising targets against KP, such as fatty acids, LPS, peptidoglycan, pyrimidine deoxyribonucleotides, and purine nucleotides, have been identified. However, designing and manufacturing inhibitors of these targets involves a multi-step process [46].

Our research indicates that, considering the local epidemiological profile of *Klebsiella pneumoniae* strains, in severe infections that are most likely due to Gram-negative bacilli, it would be reasonable to begin with combinations like aztreonam/avibactam or ceftazidime/avibactam plus aztreonam or cefiderocol according to the susceptibility profile. Although COL and FDC have similar sensitivity profiles, it would be safer to keep these options for the second-line regimen.

The most recent recommendations from the European Society of Clinical Microbiology and Infectious Diseases (ESCMID) state that, if active in vitro, CZA should be administered to patients with severe CRKP infections, except for MBL-producing strains. FDC and AZA are conditionally advised for patients with MBL-producing strains causing CRKP infections. Depending on the infection source, using older antimicrobials, such as aminoglycosides or COL, is advised for non-severe CRKP infections. For the treatment of CRKP pneumonia, TGC may be used at higher dosages, if necessary, but not for BSIs or HAP/VAP [47].

The Infectious Diseases Society of America (IDSA) suggests CZA, MEV, and IMR for patients with severe CRKP infections, excluding those of the urinary tract. CZA plus ATM is suggested for patients with a confirmed MBL-producing CRKP infection. FDC can also be used but not for urinary tract infections (UTIs). While COL should be avoided for the treatment of CRKP infections due to the associated increased mortality and severe nephrotoxicity, IDSA advises against using TGC and ERV as a monotherapy for the treatment of CRKP UTIs and BSIs [48].

## 4. Materials and Methods

### 4.1. Main Aspects

Our work was a cohort study that included prospectively and retrospectively collected data. We isolated 568 KP strains from 374 patients admitted to our hospital with various complaints between January 2023 and January 2024; 121 of these strains turned out to be CRKP (21.3%) and were taken from various anatomical sites of 71 patients associated with 77 hospital discharges. We selected 89 CRKP strains, eliminating double samples from the same site on the same person, and we further tested 45 of them for the presence of carbapenemase types (Figure 5).

### 4.2. Bacteriology

Bacterial samples of KP were obtained from various clinical sources: blood, urine, cerebrospinal fluid, sputum, tracheal secretion, bronchial aspirate, purulent discharges, abscesses, and rectal carrier screenings. These samples were cultured on Columbia agar + 5% sheep blood, MacConkey agar, CHROMID^®^ ESBL agar, and CHROMID^®^ CARBA SMART agar, all provided by bioMérieux SA (Marcy-l’Étoile, France). The cultures were incubated for 24 h at 37 °C in an aerobic environment to support typical aerobic bacterial growth. After incubation, bacterial identification was performed using the bioMérieux VITEK^®^ 2 automated system with Gram-negative (GN) identification cards. This enabled efficient identification of KP strains from various anatomical sites.

For antibiotic susceptibility testing, the bioMérieux VITEK^®^ 2 system was employed using the AST-N437, AST-N438, AST-XN24, and AST-XN26 cards. The results were interpreted based on the 2023 EUCAST clinical breakpoints for the minimal inhibitory concentration (MIC). Additionally, FDC susceptibility was determined using disk diffusion on Mueller–Hinton E agar (bioMérieux) with 30 mcg FDC disks from Oxoid Ltd. (Basingstoke, UK). Ceftazidime–avibactam and aztreonam were tested using both the VITEK^®^ 2 and disk diffusion methods.

The detection and classification of carbapenemase-producing isolates were performed using the NG-Test^®^ CARBA-5 immunochromatographic assay (NG Biotech, Guipry, France). This rapid diagnostic test allows for the identification of the five major carbapenemase enzyme families (KPC, NDM, VIM, IMP, and OXA-48-like) directly from bacterial colonies within 15 min. Importantly, the OXA-48-like enzyme family detected by this test includes multiple clinically relevant variants, such as OXA-48, OXA-181, and OXA-232.

Before testing, the isolates were subcultured twice to ensure their purity and optimal growth. Frozen stocks of isolates were initially streaked onto blood agar plates and subsequently cultured overnight on both blood agar and MacConkey agar. Following the manufacturer’s protocol, three individual colonies were collected using a 1 µL inoculation loop and resuspended in a microcentrifuge tube containing 5 drops of extraction buffer. The suspension was vortexed for approximately 3–5 s, or 10–15 s in the case of mucoid colonies, to achieve adequate homogenisation. A 100 µL aliquot of this suspension was then applied into the sample well of the NG-Test CARBA-5 device. The results were visually interpreted after 15 min, observing for the presence or absence of control and specific test lines. To minimise observer bias, the test results from colonies grown on blood and MacConkey agars were evaluated independently by different laboratory personnel.

### 4.3. Statistical Methods

We applied descriptive statistics and report the range, mean, and standard deviation (s.d.) to characterise the main aspects related to samples, discharges, and patients. We also emphasise the incidence and co-existence of different types of carbapenemases. The Fisher Exact Test was applied to assess the associations between different resistance enzymes and the susceptibility profile.

The same test was used to examine the associations between patient demographic profiles, hospitalisation characteristics, and outcomes. Furthermore, two logit models were specified to estimate the probability of death during a hospital stay and the probability of sepsis developing following infection. We introduced several covariates into these models, such as intensive care unit (ICU) stays, the length of hospital stays (LoS), and the occurrence of surgical intervention. For each model, we computed and reported the odd ratios (aORs) and the correspondent 95% confidence intervals (95%CIs). Both logit models were validated using the Hosmer–Lemeshow test (HL) [49], Nagelkerke R squared, and overall correct classification. For each test, we used *p*-value (*p*) = 0.05 as the threshold for significance.

## 5. Limitations

The limitations of our study include the single-centre design and its retrospective nature. In addition, some of the subgroup analyses were based on a small number of isolates, affecting the robustness and statistical reliability of certain results.

Carbapenemase detection was only performed on a subset of 45 out of 89 CRKP isolates, as the NG-Test^®^ Carba 5 assay was not available at the time of initial isolate processing. Furthermore, antimicrobial susceptibility testing was not fully consistent across all 89 isolates due to variability in VITEK 2 AST card availability.

In addition, it is worth mentioning that molecular typing methods, such as multi-locus sequence typing (MLST) and whole genome sequencing (WGS) would have significantly strengthened the findings and provided deeper insight into the clonal relatedness and genetic determinants of resistance.

Another limitation of this study concerns the disk diffusion (DD) method used for cefiderocol, that could provide false positive results regarding the lack of susceptibility. However, DD test results showing susceptibility to FDC are reliable. Yet, the broth microdilution method (BMD) is recommended for results within the resistance zone or ATU. BMD is the EUCAST—recommended method and, in subsequent internal validation efforts within our laboratory, we recently implemented BMD-based testing of cefiderocol and other last-resort agents.

Due to these limitations and the particularities in resistance enzymatic load of CRKP, the local epidemiological profile does not allow applicability to other regional medical care facilities.

## 6. Conclusions

Epidemiological analysis of our cohort demonstrated that MBL-producing KP strains, predominantly co-harboring NDM/ and OXA-48-like enzymes, represented the majority of the CRKP isolates, thereby compromising most currently available therapeutic options. Evaluation of FDC activity using the disk diffusion method raised concerns regarding potential suboptimal susceptibility, despite the fact that this agent was scarcely used in Romania during the study period.

Prolonged hospitalization and ICU admission were identified as the main risk factors significantly associated with increased rates of sepsis and mortality, whereas undergoing surgical procedures correlated with a reduced risk of these outcomes.

The development of novel antimicrobial agents, in conjunction with the reinforcement of antimicrobial stewardship and infection prevention strategies, remains essential for the management of infections caused by MBL-producing CRKP. Rapid molecular diagnostic tools represent an effective approach to promptly identify both the pathogen and its resistance mechanisms, facilitating the early initiation of tailored effective therapy. 

Future efforts should focus on incorporating comprehensive susceptibility data for FDC, ATM/AZT, and other emerging antibiotics into antimicrobial stewardship programs to optimize treatment outcomes. 

## Figures and Tables

**Figure 1 antibiotics-14-00783-f001:**
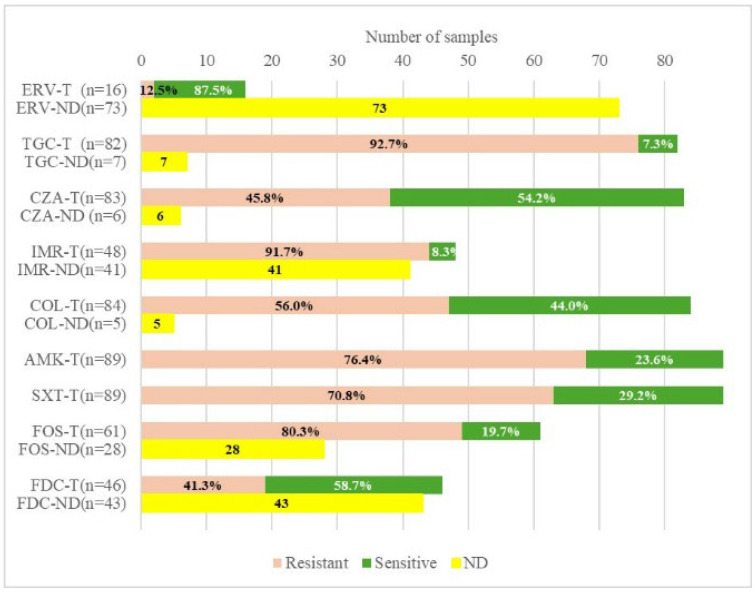
Antibiotic susceptibility for selected strains. ERV—eravacycline; TGC—tigecycline; CZA—ceftazidime/avibactam; IMR—imipenem/relebactam; COL—colistin; AMK—amikacin; SXT—trimethoprim-sulfamethoxazole; FOS—fosfomycin; FDC—cefiderocol; T—tested; ND—not determined.

**Figure 2 antibiotics-14-00783-f002:**
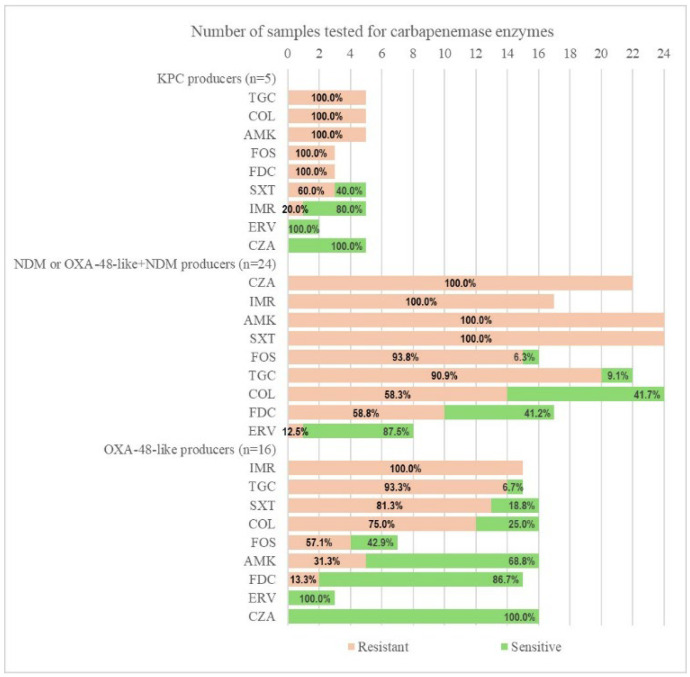
Antibiotic susceptibility according to the type of carbapenemase identified in selected strains. ERV—eravacycline; TGC—tigecycline; CZA—ceftazidime/avibactam; IMR—imipenem/relebactam; COL—colistin; AMK—amikacin; SXT—trimethoprim-sulfamethoxazole; FOS—fosfomycin; FDC—cefiderocol.

**Figure 3 antibiotics-14-00783-f003:**
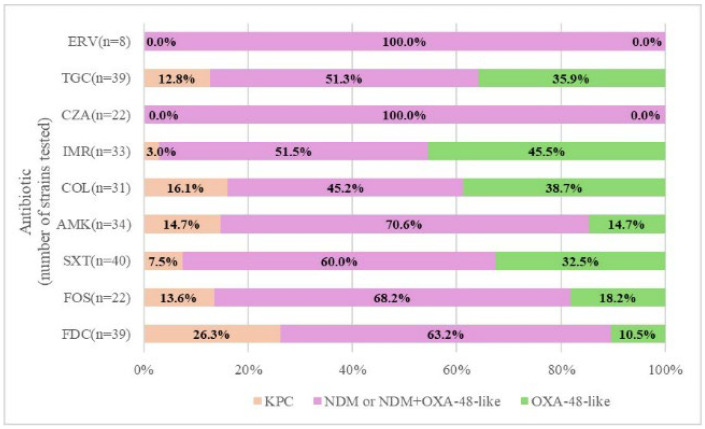
Antibiotic resistance and enzymatic profile in tested strains. ERV—eravacycline; TGC—tigecycline; CZA—ceftazidime/avibactam; IMR—imipenem/relebactam; COL—colistin; AMK—amikacin; SXT—trimethoprim-sulfamethoxazole; FOS—fosfomycin; FDC—cefiderocol.

**Figure 4 antibiotics-14-00783-f004:**
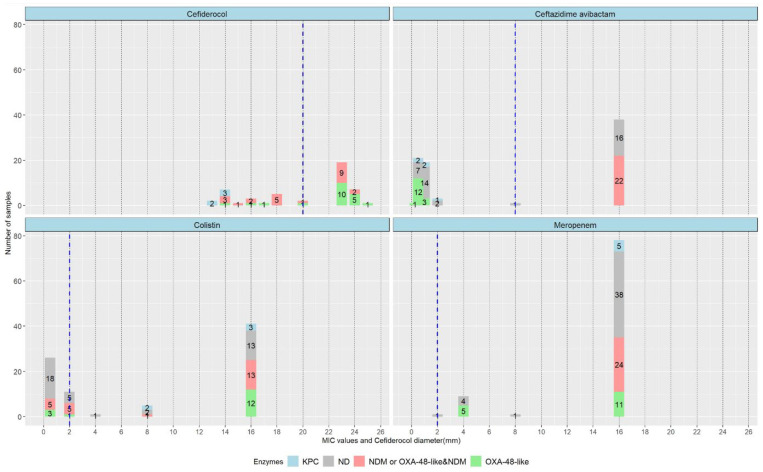
Antibiotic MIC/diameter values, breakpoints and strain distribution correlated with enzymatic profile, in tested strains. Carbapenemases—KPC, OXA-48-like, NDM; ND—not determined.

**Figure 5 antibiotics-14-00783-f005:**
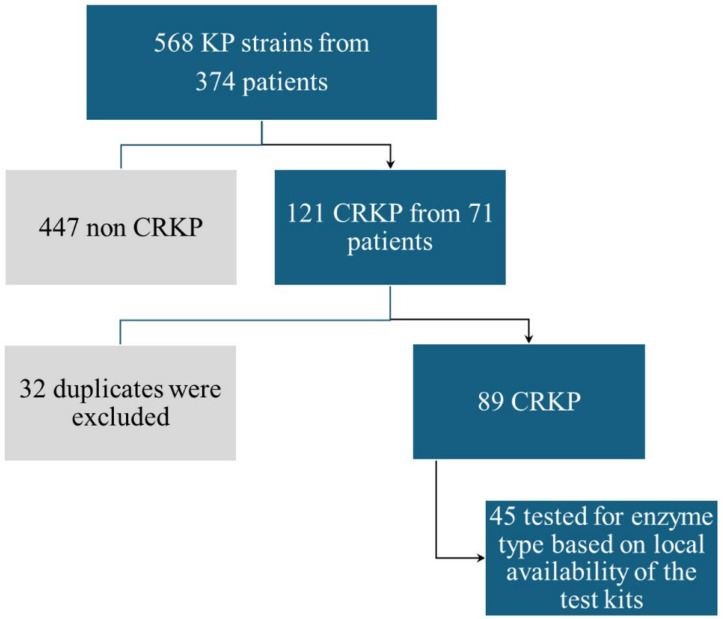
Algorithm criteria for selected strains. KP—*Klebsiella pneumoniae*; CRKP—carbapenem-resistant *Klebsiella pneumoniae*.

**Table 1 antibiotics-14-00783-t001:** Samples and hospital events profiles, treatment and outcomes.

Factors	Values	Samples		Hospital Events (Discharges)	
		n = 89	% of Total Valid Cases	n = 77	% of Total Valid Cases
Gender	Males	67	75.3	59	76.6
	Females	22	24.7	18	23.4
Ward	Surgical	44	49.4	41	53.3
	Medical	45	50.6	36	46.7
Origin	Community	22	24.7	21	27.3
	Hospital	67	75.3	56	72.7
Charlson score	Below 5	36	41.9	28	37.8
	At least 5	50	58.1	46	62.2
Events with Intensive Care Unit (ICU) stay	Yes	-	-	45	58.4
	No	-	-	32	41.6
Surgical intervention	Yes	-	-	43	55.8
	No	-	-	34	44.2
Sepsis	Yes	32	36.0	24	31.2
	No	57	64.0	53	68.8
Sample site	Abdominal	3	3.4	3	3.9
	Blood culture	5	5.6	5	6.5
	Wound	8	9.0	8	10.4
	Respiratory tract	13	14.6	13	16.9
	Urinary tract	32	36.0	32	41.6
	Others	13	14.5	13	16.8
Active treatment	Yes	50	56.2	46	59.7
	No	39	43.8	31	40.3
Patients with active treatment	Yes	50	56.2	46	59.7
	No	39	43.8	31	40.3
Current portage	Yes	-	-	34	44.7
	No	-	-	42	55.3
Portage and sepsis	Yes	-	-	15	19.7
Fatality during hosp. stay	Yes	-	-	15	19.5
	No	-	-	62	81.5
Covariates	Statistics		Samples (days)	Hospital Events (Discharges) (days)	
Age in years	mean (s.d.)		62.5 (14.0)	63.7 (13.8)	
Length of hospital stay (LoS)	mean (s.d.)		-	23.2 (19.1)	
ICU stay, excluding non-ICU cases	mean (s.d.)		-	12.2 (11.1)	
Active treatment	mean (s.d.)		-	13.6 (13.4)	
Time from admission to sample collection	mean (s.d.)		9.1 (12.4)	-	

**Table 2 antibiotics-14-00783-t002:** Association between different patient treatment events.

	ICU Stay	Charlson Group	Surgical Intervention	Sepsis	Active Treatment
ICU stay	1	0.200 (*p* = 0.087)	0.144 (*p* = 0.210)	0.407 (*p* < 0.001)	0.160 (*p* = 0.164)
Charlson group		1	0.163 (*p* = 0.162)	0.064 (*p* = 0.580)	−0.034 (*p* = 0.769)
Surgical intervention			1	−0.079 (*p* = 0.487)	0.283 (*p* = 0.013)
Sepsis				1	0.267 (*p* = 0.019)
Active treatment					1

**Table 3 antibiotics-14-00783-t003:** Sepsis cumulative risk, probability of death during hospitalisation and associated causes.

Covariates (Factors)	Model			
	Mortality		Sepsis	
	aOR (95% CI)	*p*-Value	aOR (95% CI)	*p*-Value
LoS (days)	1.00 (0.95–1.05)	0.919	1.03(1.00–1.07)	0.048
ICU stay (ref = no stay)	40.8(3.5–473.3)	*p* < 0.01	10.0(2.4–41.0)	*p* < 0.01
Surgical intervention (ref = no intervention)	0.05(0.01–0.33)	*p* < 0.01	0.26(0.07–0.97)	0.045
Charlson group (ref = below 5)	4.8(0.74–30.9)	0.101	1.4(0.39–4.8)	0.617
Constant	0.02	*p* < 0.01	0.07	*p* < 0.01
Sample size	73		73	
Nagelkerke R squared	0.47		0.33	
HL-test	*p* = 0.93		*p* = 0.79	
Overall correct classification	84.9%		76.7%	

## Data Availability

Data are contained within the article.

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
