# Peer review of "Epidemiology of Carbapenem-Resistant Klebsiella Pneumoniae Co-Producing MBL and OXA-48-Like in a Romanian Tertiary Hospital: A Call to Action"

_antibiotics, 2025, doi:10.3390/antibiotics14080783_

Round 1
Reviewer 1 Report
Comments and Suggestions for Authors
Dear Editor The paper entitled "Epidemiology of Carbapenem-Resistant Klebsiella pneumoniae Co-producing MBL and OXA-48-like in a Romanian Tertiary Hospital: a call to action" is very interesting, however, the authors could have confirmed that the results were performed using only the NG-Test® 146 CARBA-5 immunochromatographic assay. The authors could improve the study by confirming the results using another phenotypic test or even confirming them by PCR reaction.
Author Response
Comments 1: The authors could improve the study by confirming the results using another phenotypic test or even confirming them by PCR reaction.
Response 1: Thank you for pointing this out. We did not use another phenotypic test in this study and we will keep in mind your suggestion for our future projects. Four strains tested through NG-Test 146 Carba-5 assay were confirmed by PCR methods as having the specified carbapenemases.
Reviewer 2 Report
Comments and Suggestions for Authors
The manuscript explores the epidemiology of carbapenem-resistant KP in Romanian tertiary hospital.
The relevant tables must be included in the main manuscript rather than the supplementary files.
The authors refer to Figures 1,2,3,4 etc., however, there are no figures included in the manuscript. Please add relevant figures.
In section 3.2, please clarify if only 45 out of 89 samples were examined for carbapenemase production or did 45 tested positive? What about the resistance profile of remaining 44?
Line 302: needs consistent formatting
The surgical intervention data in relation to sepsis is too small. Please elaborate.
Please include the limitations of this study.
Author Response
Comments 1: The relevant tables must be included in the main manuscript rather than the supplementary files.
Response 1: Thank you for pointed this out. We have added them in the manuscript.
Comments 2: The authors refer to Figures 1,2,3,4 etc., however, there are no figures included in the manuscript. Please add relevant figures.
Response 2: We agree with this comment. Therefore, we have added them in the manuscript.
Comments 3: In section 3.2, please clarify if only 45 out of 89 samples were examined for carbapenemase production or did 45 tested positive? What about the resistance profile of remaining 44?
Response 3: We thank the reviewer for this pertinent observation. At the time of initial testing, we did not have access to the NG-Test® Carba 5 (NG Biotech, France), a lateral flow immunochromatographic assay used for the rapid identification of the five main carbapenemase families (KPC, OXA-48-like, VIM, IMP, and NDM). As a result, only the subset of 45 isolates tested after the assay became available could be phenotypically characterised for carbapenemase production.
Unfortunately, by the time we acquired the necessary test kits, the remaining 44 CRKP isolates were no longer viable due to storage limitations, and thus we were unable to complete either the carbapenemase detection or the full antimicrobial susceptibility testing on those strains.
Comments 4: Line 302: needs consistent formatting
Response 4: Line 302 was formatting.
Comments 5: The surgical intervention data in relation to sepsis is too small. Please elaborate.
Response 5: The surgical intervention data in relation to sepsis is elaborated in lines 390-403.
Comments 6: Please include the limitations of this study.
Response 6: Limitations are elaborated in lines 592-605.
Reviewer 3 Report
Comments and Suggestions for Authors
Dear authors, Epidemiology of Carbapenem-Resistant Klebsiella pneumoniae Co-producing MBL and OXA-48-like in a Romanian Tertiary Hospital: a call to action.” After reading your manuscript, I have some comments.
The manuscript has no abstract.
Please review the use of abbreviations; some, such as PK and PD, are not used again in the text.
Among the 89 CRKP isolates, carbapenemase types were identified in only 45 isolates, which was the selection criterion.
Paragraphs 234-246: Why was susceptibility testing to all drugs not conducted for all strains?
In all figures, it is necessary to add the meanings of all abbreviations to the figure captions.
Supplementary tables must be in the article
Tables where associations and risk factors are presented should include the total number of patients across groups, the number of patients with and without sepsis, and the number of patients who died and survived.
Author Response
Comments 1: The manuscript has no abstract.
Response 1: We do not understand the reason you do not have access to the abstract. The abstract was submitted accordingly.
Comments 2: Please review the use of abbreviations; some, such as PK and PD, are not used again in the text.
Response 2: Thank you for pointing this out. The abbreviations were revised.
Comments 3: Among the 89 CRKP isolates, carbapenemase types were identified in only 45 isolates, which was the selection criterion.
Response 3: We enrolled every consecutive strain that was identified in the lab and showed resistance to carbapenems, excluding duplicates. We did not track CRKP strain clones, and indeed, that might contain potential bias in strain selection. We acknowledge that this is a potential limitation of the study. Only 45 of them were tested due to our laboratory availability of NG- tested Carba -5 assay.
Comments 4: Paragraphs 234-246: Why was susceptibility testing to all drugs not conducted for all strains?
Response 4: We thank the reviewer for this pertinent observation. Antimicrobial susceptibility testing among the 89 isolates was not fully uniform, as some strains were tested against slightly different panels of antibiotics. This variation resulted from the use of different VITEK 2 AST cards, depending on reagent availability at the time of testing. As the composition of antibiotics varies between card types, not all isolates were tested against the exact same set of agents. We acknowledge this as a limitation and have clarified it accordingly in the revised manuscript.
We have now clarified this constraint in the Limitations section of the manuscript to ensure full transparency.
Comments 5: In all figures, it is necessary to add the meanings of all abbreviations to the figure captions.
Response 5: We revised this issue and we have added all the meanings and abbreviations to the figure captions.
Comments 6: Supplementary tables must be in the article. Tables where associations and risk factors are presented should include the total number of patients across groups, the number of patients with and without sepsis, and the number of patients who died and survived.
Response 6: We have added the tables to the manuscript body and modified Table 1 accordingly.
Reviewer 4 Report
Comments and Suggestions for Authors
In the present study, the authors aimed to describe the epidemiological characteristics of the CRKP strains in a multidisciplinary emergency hospital in Romania, especially those producing MBLs with or without other carbapenemases. In addition, they have determined the characteristics of the patient who may be infected and/or colonized with CRKP, as well as to assess susceptibility to available antibiotics and clinical outcome of patients infected with these strains. The manuscript needs extensive editing in English from a native speaker. Please use the past tense when describing the methods used and results. The methodology used should be improved, as recommended below.
Major comments
- As the authors state in the limitations of the study (lines 574-575), it consists a single centre designed study, retrospective nature and some potential biases in data collection and analysis and they have pointed some results in a small number of isolates, with limited statistical reliability. Moreover, The local epidemiological profile does not allow applicability in other regional medical care facilities. The authors should use molecular typing methods, such as MLST and WGS so as to characterise the strains and their resistance mechanisms, so as to overcome all these issues.
- Lines 194-196: please move these linesto the end of the introduction section.
- Lines 494-495: The authors found that we found that 41.3% of the tested strains were 494 non-susceptible to FDC by disk diffusion method. The MIC of cefiderocol for all vital isolates should also be determined using broth microdilution (please see: Kriz, R., Spettel, K., Pichler, A. et al. In vitro resistance development gives insights into molecular resistance mechanisms against cefiderocol. J Antibiot 77, 757–767 (2024). https://doi.org/10.1038/s41429-024-00762-y).
The English could be improved to more clearly express the research.
Author Response
Comments 1: The authors should use molecular typing methods, such as MLST and WGS so as to characterise the strains and their resistance mechanisms, so as to overcome all these issues.
Response 1: We fully acknowledge the reviewer’s important observation regarding the limitations posed by the single-centre, retrospective nature of our study and the relatively small number of isolates in some subgroups. We agree that molecular typing methods, such as MLST and whole genome sequencing (WGS), would have significantly strengthened the findings and provided deeper insight into the clonal relatedness and genetic determinants of resistance.
Comments 2: Lines 194-196: please move these lines to the end of the introduction section.
Response 2: Lines 194-196 were moved.
Comments 3: Lines 494-495: The authors found that we found that 41.3% of the tested strains were non-susceptible to FDC by disk diffusion method. The MIC of cefiderocol for all vital isolates should also be determined using broth microdilution (please see: Kriz, R., Spettel, K., Pichler, A. et al. In vitro resistance development gives insights into molecular resistance mechanisms against cefiderocol. J Antibiot 77, 757–767 (2024). https://doi.org/10.1038/s41429-024-00762-y).
Response 3: We thank the reviewer for this valuable comment and for pointing us to the relevant recent publication by Kriz et al. (2024), which provides important insights into cefiderocol resistance mechanisms and reinforces the recommendation of broth microdilution (BMD) as the reference method for determining cefiderocol MICs.
In our study, we acknowledge that cefiderocol susceptibility was assessed using disk diffusion due to limited access to commercial BMD panels at the time of testing. We are aware of the limitations of disk diffusion for cefiderocol and have addressed this in the manuscript (see Discussion section). As correctly highlighted, BMD is the EUCAST-recommended method, and in subsequent internal validation efforts within our laboratory, we have implemented the Micronaut system for BMD-based testing of cefiderocol and other last-resort agents.
Round 2
Reviewer 2 Report
Comments and Suggestions for Authors
The authors have addressed all the comments that has improved the readability of the manuscript.
Author Response
Thank you for your pertinent suggestions.
Reviewer 3 Report
Comments and Suggestions for Authors
The authors have made the suggested changes and the manuscript has improved
Author Response
Thank you for your pertinent suggestions.
The entire manuscript supported extensive English editing by MDPI team.
Reviewer 4 Report
Comments and Suggestions for Authors
Please include the following responses to reviwers' comments in the paragraph of the limitations of the study:
Response 1: We fully acknowledge the reviewer’s important observation regarding the limitations posed by the single-centre, retrospective nature of our study and the relatively small number of isolates in some subgroups. We agree that molecular typing methods, such as MLST and whole genome sequencing (WGS), would have significantly strengthened the findings and provided deeper insight into the clonal relatedness and genetic determinants of resistance.
2. In our study, we acknowledge that cefiderocol susceptibility was assessed using disk diffusion due to limited access to commercial BMD panels at the time of testing. We are aware of the limitations of disk diffusion for cefiderocol and have addressed this in the manuscript (see Discussion section). As correctly highlighted, BMD is the EUCAST-recommended method, and in subsequent internal validation efforts within our laboratory, we have implemented the Micronaut system for BMD-based testing of cefiderocol and other last-resort agents.
Author Response
Comments: Please include the following responses to reviewers' comments in the paragraph of the limitations of the study.
Response: Thank you for your suggestions. We included paragraph 1 in 613-617 lines and 624-628 lines. Also, paragraph 2 is added in 629-638 lines.